# Structure of a bacterial Rhs effector exported by the type VI secretion system

Patrick Günther[1]�, Dennis Quentin[1]�, Shehryar Ahmad[2], Kartik Sachar[2], Christos Gatsogiannis[1]¤, John C. Whitney[2,3,4]*, Stefan Raunser[1]*

1 Department of Structural Biochemistry, Max Planck Institute of Molecular Physiology, Dortmund, Germany, 2 Department of Biochemistry and Biomedical Sciences, McMaster University, Hamilton, Canada, 3 Michael DeGroote Institute for Infectious Disease Research, McMaster University, Hamilton, Canada, 4 David Braley Centre for Antibiotic Discovery, McMaster University, Hamilton, Canada

☽ These authors contributed equally to this work.
¤ Current address: Institute for Medical Physics and Biophysics and Center for Soft Nanoscience, Westfälische Wilhelms-Universität Münster, Münster, Germany
* jwhitney@mcmaster.ca (J.C.W.); stefan.raunser@mpi-dortmund.mpg.de (S.R.)

**Data Availability Statement:** The cryo-EM maps of VgrG1 and RhsA have been deposited in the Electron Microscopy Data Bank (EMDB) under the accession codes of EMD-13843 and EMD-13867,

## Abstract

The type VI secretion system (T6SS) is a widespread protein export apparatus found in Gram-negative bacteria. The majority of T6SSs deliver toxic effector proteins into competitor bacteria. Yet, the structure, function, and activation of many of these effectors remains poorly understood. Here, we present the structures of the T6SS effector RhsA from *Pseudomonas protegens* and its cognate T6SS spike protein, VgrG1, at 3.3 Å resolution. The structures reveal that the rearrangement hotspot (Rhs) repeats of RhsA assemble into a closed anticlockwise β-barrel spiral similar to that found in bacterial insecticidal Tc toxins and in metazoan teneurin proteins. We find that the C-terminal toxin domain of RhsA is autoproteolytically cleaved but remains inside the Rhs 'cocoon' where, with the exception of three ordered structural elements, most of the toxin is disordered. The N-terminal 'plug' domain is unique to T6SS Rhs proteins and resembles a champagne cork that seals the Rhs cocoon at one end while also mediating interactions with VgrG1. Interestingly, this domain is also autoproteolytically cleaved inside the cocoon but remains associated with it. We propose that mechanical force is required to remove the cleaved part of the plug, resulting in the release of the toxin domain as it is delivered into a susceptible bacterial cell by the T6SS.

## Author summary

Bacteria have developed a variety of strategies to compete for nutrients and limited resources. One system widely used by Gram-negative bacteria is the T6 secretion system which delivers a plethora of effectors into competing bacterial cells. Known functions of effectors are degradation of the cell wall, the depletion of essential metabolites such as NAD$^+$ or the cleavage of DNA. RhsA is an effector from the widespread plant-protecting bacteria *Pseudomonas protegens*. We found that RhsA forms a closed cocoon similar to that found in bacterial Tc toxins and metazoan teneurin proteins. The effector cleaves its polypeptide chain by itself in three pieces, namely the N-terminal domain including a

respectively. The refined models for VgrG1 and RhsA were uploaded in the PDB and the entries have the IDs 7Q5P and 7Q97, respectively. The original data of the de novo protein sequencing is available inprovided as supplementary data (the S1 Dataset).

**Funding:** This work was supported by the Max Planck Society (to S.R.) and a Discovery Grant from the Natural Sciences and Engineering Research Council of Canada (RGPIN-2017–05350 to J.C.W.). J.C.W. is the Canada Research Chair in Molecular Microbiology and holds an Investigators in the Pathogenesis of Infectious Disease Award from the Burroughs Wellcome Fund. The funders had no role in study design, data collection and analysis, decision to publish, or preparation of the manuscript.

**Competing interests:** The authors have declared that no competing interests exist.

seal, the cocoon and the actual toxic component which potentially cleaves DNA. The toxic component is encapsulated in the large cocoon, so that the effector producing bacterium is protected from the toxin. In order for the toxin to exit the cocoon, we propose that the seal, which closes the cocoon at one end, is removed by mechanical forces during injection of the effector by the T6 secretion system. We further hypothesize about different scenarios for the delivery of the toxin into the cytoplasm of the host cell. Together, our findings expand the knowledge of the mechanism of action of the T6 secretion system and its essential role in interbacterial competition.

## Introduction

One way that bacteria interact with their environment is by secreting toxic molecules into their surroundings. Many species of Gram-negative bacteria have evolved specialized protein secretion systems for this purpose with arguably the best-characterized example being the T6SS [1,2]. T6SSs mediate bacterial antagonism by facilitating the injection of toxic effector proteins into competing bacterial cells in a contact-dependent manner [3].

A functional T6SS apparatus requires the concerted action of three protein subcomplexes: a cell envelope-spanning membrane complex, a cytoplasmic baseplate complex and a bacteriophage tail-like sheath-tube complex [4,5]. The tail-like complex consists of a contractile sheath that surrounds an inner tube comprised of many copies of stacked ring-shaped hexameric hemolysin co-regulated protein (Hcp) [6,7]. This Hcp tube is capped with a member of the homotrimeric valine-glycine repeat protein G (VgrG) spike protein family, which typically also interacts with a cone-shaped proline-alanine-alanine-arginine (PAAR) domain-containing protein to form a complete T6SS tail tube-spike complex [8]. During a T6SS firing event, the Hcp-VgrG-PAAR tube-spike complex is rapidly assembled in the cytosol of the attacking bacterium, recruited to the membrane complex through its interaction with the baseplate, and subsequently surrounded by a contractile sheath that upon sheath contraction, propels the tail spike into the recipient cell [9]. Toxic effector proteins are recruited to the tube-spike complex and injected into recipient cells alongside Hcp and VgrG, typically resulting in death of the target cell. Self-protection from antibacterial T6SS effectors is accomplished via effector co-expression with cognate immunity proteins, which neutralize effector toxicity by occluding the effector active site or by the hydrolysis of effector generated toxic products [10,11].

Effectors are recruited to distinct regions of the T6SS tube-spike complex. Smaller effectors (< 50 kDa) are typically recognized and loaded within the lumen of the Hcp tube [12] whereas larger, multidomain effectors interact with VgrG proteins via an effector-encoded PAAR domain or by directly interacting with VgrG itself [13]. PAAR-containing effectors are widely distributed in Proteobacteria and can be further subdivided into subfamilies of proteins based on the presence of additional sequence motifs that may play a role in effector translocation into target cells. One family of PAAR effectors that was recently described in detail possess an additional N-terminal motif, called prePAAR, predicted to be involved in PAAR folding, as well as at least one N-terminal transmembrane domain (TMD) that is hypothesized to insert into the recipient cell inner membrane with one to three transmembrane helices per TMD that enable toxin translocation into the cytoplasm [14]. TMD-containing PAAR effectors require effector-associated gene (Eag) chaperones to bind and stabilize effector TMDs prior to export from the donor cell [14,15].

Previously, our group showed that the T6SS of the soil bacterium *Pseudomonas protegens* secretes the prePAAR and PAAR-containing effector RhsA [14]. The *rhsA* operon contains

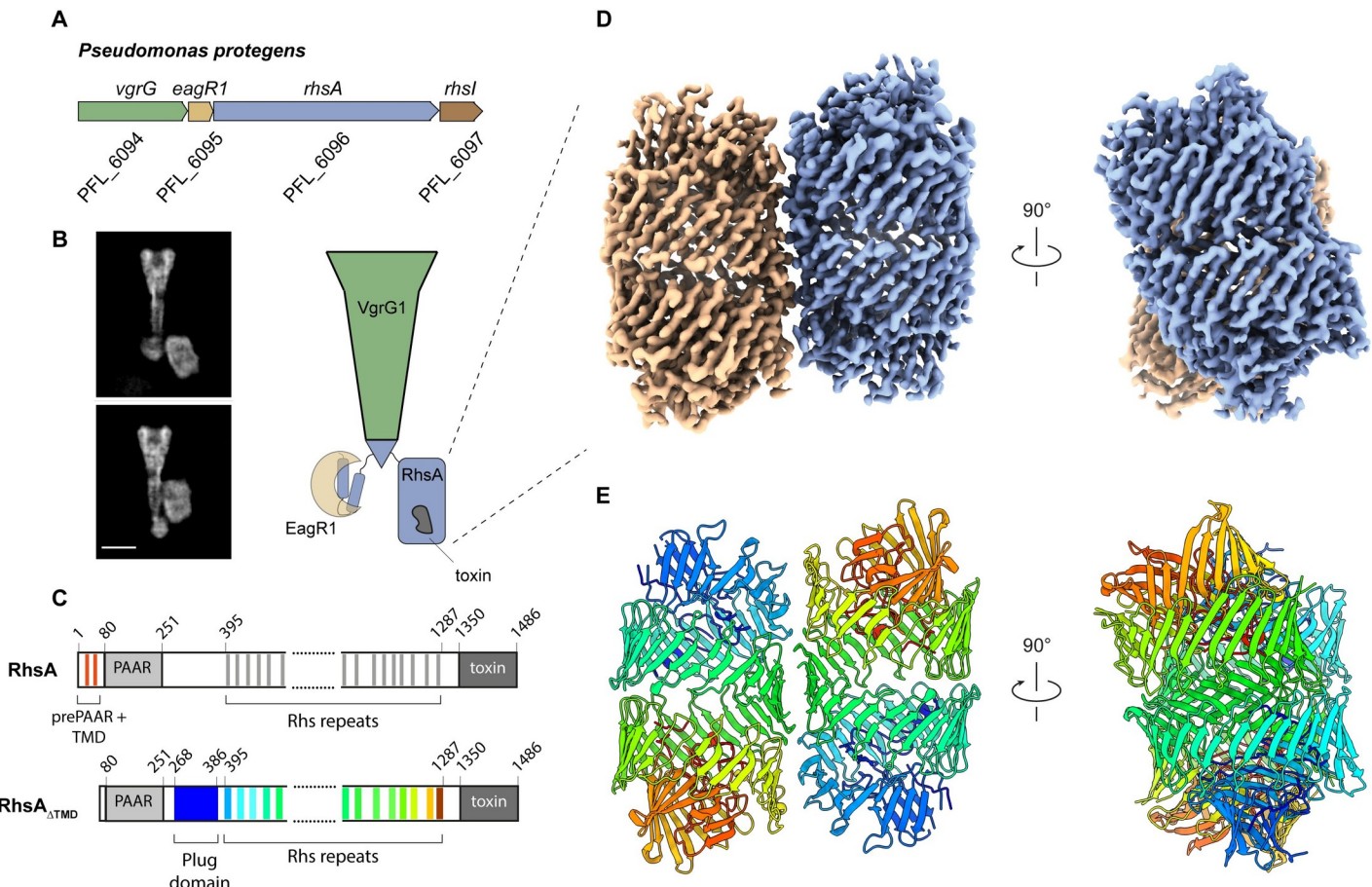

**Fig 1. High-resolution structure of the T6SS effector RhsA.** (A) Genomic context of *rhsA* (PFL_6096, blue) in *Pseudomonas protegens* Pf-5. Upstream genes encoding the cognate VgrG protein, *vgrG1* (PFL_6094, green) and RhsA-specific chaperone, *eagR1* (PFL_6095, light brown) are shown. Self-protection against RhsA is conferred via expression of the downstream immunity-encoding gene, *rhsI* (dark brown). (B) Representative cryo-EM 2-D class averages of the assembled pre-firing complex composed of VgrG1, RhsA and EagR1 (left) and a schematic representation of each of the components that comprise this complex (right). Scale bar, 10 nm. (C) Full-length RhsA contains a prePAAR motif and a TMD comprising two transmembrane helices upstream of its PAAR domain. The RhsA$_{\Delta TMD}$ truncation of RhsA was used in this study to determine the high-resolution structure of the RhsA cocoon. (D) Cryo-EM density of RhsA$_{\Delta TMD}$ displayed perpendicular to the central symmetry axis of the barrel and rotated 90° clockwise (map postprocessed with DeepEMhancer). (E) Cartoon representation of the atomic model of RhsA colored in rainbow from N-terminus (blue) to C-terminus (red).

genes encoding its cognate VgrG spike and Eag chaperone proteins (Fig 1A), both of which are required for the delivery of this effector into susceptible competitors [14]. Previous negative-stain electron microscopy experiments suggest that EagR1, RhsA and VgrG assemble to form a complex necessary for T6SS function (Fig 1B) [14]. The N-terminus of RhsA consists of a prePAAR motif as well as two predicted transmembrane helices within its TMD. The pre-PAAR motif is proposed to contribute sequence elements that 'complete' the PAAR domain and enable its interaction with the tip of VgrG1 [14] whereas the transmembrane helices likely play a role in target cell penetration [16].

In addition to its N-terminal prePAAR, TMD and PAAR regions, RhsA possesses Rhs elements, which are characterized by the presence of repeating tyrosine-aspartate (YD) motifs. The chromosomal regions encoding Rhs repeats were originally described in *Escherichia coli* genomes as sites where recombination frequently took place [17]. Later studies revealed that Rhs repeat-containing proteins are not unique to *E. coli* but are common across Proteobacteria [18,19]. Other more distant homologs, called teneurins, are involved in axon guidance in

vertebrates [20–22]. Teneurins are type II single-pass transmembrane proteins that mediate cell-cell adhesion via their Rhs repeat-containing extracellular domains [23,24]. The intracellular domain consists of a transcriptional repressor that is released from the membrane upon proteolytic cleavage. The extracellular epidermal growth factor-like repeats (EGF) mediate covalent dimerization by formation of disulphide bridges [21,25]. Structures of bacterial Rhs proteins and the extracellular domain of human teneurin2 revealed that Rhs repeat regions form a large (~100 kDa) β-barrel cocoon-like structure spirally wound counterclockwise [26–29]. The C-terminus of the Rhs barrel is typically demarcated by a highly conserved PxxxxDPxGL motif, which is necessary for autoproteolytic cleavage [20]. Previous work has shown that C-terminal autoproteolysis of both T6SS Rhs effectors and toxin complex (Tc) Rhs toxins is required for toxin release from the Rhs cocoon [26,27,30].

Tc toxins consist of three components: a 1.4 MDa homopentameric A component (TcA), a ~170 kDa TcB subunit and a ~100 kDa TcC subunit [26]. TcA acts as membrane permeation and translocation device, while TcB and TcC together form a large hollow cocoon composed of Rhs repeats. The C-terminal extension of the TcC subunit lies inside the cocoon and is autocatalytically processed by an aspartyl protease [26]. While the Rhs repeat-containing region is highly conserved among T6SS Rhs effectors and Tc toxins, their N- and C-terminal extensions can differ significantly. The C-terminus of bacterial Rhs proteins encode for a diverse range of toxins and is often referred to as the hypervariable region (HVR) [31]. In Tc toxins, the C-terminal toxin is encapsulated within the Rhs cocoon [32], however, previous structural analyses of these proteins have been unable to resolve the structure of a HVR, suggesting that this region may be partially unfolded. So far, no high-resolution structural information exists for T6SS Rhs effectors.

The Rhs repeat-containing cocoon TcBC interacts through the β-propeller domain of TcB with TcA. The propeller acts as a gate that opens upon A component binding, allowing the HVR to enter the translocation channel of TcA [32]. The penetration of the target cell membrane through the A component is triggered by either a shift to higher or lower pH [26]. Membrane penetration opens the translocation channel of TcA, which permits subsequent HVR release into the prey cell cytosol [26,33]. A similarly detailed toxin release mechanism has not been reported for the bacteria targeting T6SS Rhs effectors [3] and structures of T6SS-exported Rhs proteins have not been determined to date.

In this work, we present high-resolution structures of the T6SS effector RhsA and its cognate VgrG1 protein at resolutions of 3.3 Å. The structure of VgrG1 is nearly identical to our previously determined VgrG1 structure from *P. aeruginosa* [16], indicating a high structural conservation of VgrG proteins. The structure of RhsA reveals that the Rhs-repeats of RhsA form a cocoon similar to that of the B and C components of Tc toxins and teneurins. However, unlike the cocoon of Tc toxins, which are capped by the aforementioned β-propeller domain, RhsA is closed at its N-terminus by a plug domain that connects the cocoon to its N-terminal PAAR domain. The plug domain is held in place via hydrophobic interactions with the Rhs cocoon and its eventual opening is the most likely event for the toxin domain to be released from the barrel. Our structural and biochemical findings show that the plug domain is autoproteolytically cleaved inside the cocoon. The cleaved 304 N-terminal residues, comprising the prePAAR motif, TMD, and PAAR domain also include an N-terminal seal peptide of the Rhs plug and an anchor helix. This helix interacts extensively with the inner surface of the cocoon and is likely responsible for keeping the cleaved N-terminal 304 residues physically associated with the cocoon. In addition, we find that the RhsA C-terminal toxin domain is also cleaved via a canonical Rhs aspartyl autoprotease and that this cleavage event is required for RhsA-dependent bacterial killing. Taken together, our results provide novel insights into the architecture and mechanism of action of Rhs effectors exported by the T6SS.

## Results and discussion

### RhsA forms a cocoon-like structure that undergoes N- and C-terminal autocleavage

We previously demonstrated that a fragment of RhsA lacking its N-terminal prePAAR motif and TMD (Fig 1C, residues 74–1486), RhsA$_{\Delta TMD}$, is stable in the absence of its EagR1 chaperone and can be purified to homogeneity as a soluble protein when over-expressed in *Escherichia coli* [14]. We therefore expressed and purified RhsA$_{\Delta TMD}$ (S1A Fig) and used it for cryo-EM and single particle analysis. The raw images and 2D class averages suggest that RhsA$_{\Delta TMD}$ forms a stable dimer in solution, which facilitated structure determination due to the increased size of the particles. Subsequent image processing imposing $C_2$ symmetry resulted in a reconstruction with a resolution of 3.3 Å from 454,740 particles (Fig 1D and Table 1). The high quality of the map allowed us to build an atomic model of 75% of the protein, comprising residues 268–275, 289–302, 305–366, 371–386, 395–871, 891–1039 and 1051–1350 (Fig 1E). The 25% of remaining unresolved regions correspond to the PAAR domain (residues 75–267) and the

**Table 1. Statistics of cryo-EM data collection, image processing and model validation.**

|  | VgrG1 | RhsA$_{\Delta TMD}$ |
| --- | --- | --- |
| **Data collection** |  |  |
| Microscope | Titan Krios (X-FEG, Cs-corrected) | Titan Krios (X-FEG, Cs 2.7 mm) |
| Magnification | 59,000 | 105,000 |
| Voltage (kV) | 300 | 300 |
| Defocus range (μm) | -1.2 to -2.2 | -0.8 to -2.2 |
| Camera | F3 linear | K3 counting |
| Pixel size (Å/pixel) | 1.1 | 0.91 |
| Total electron dose (e$^-$/Å$^2$) | 90 | 62 |
| Exposure time (s) | 1.5 | 2 |
| Number of images | 1250 | 13,090 |
| **Refinement** |  |  |
| Number of final particles | 423,980 | 454,740 |
| Final resolution (Å) | 3.3 | 3.3 |
| Symmetry | $C_3$ | $C_2$ |
| Map sharpening B factor (Å$^2$) | -174.5 | DeepEMhancer |
| **Model composition** |  |  |
| Non-hydrogen atoms | 14868 | 17070 |
| Protein residues | 1881 | 2094 |
| RMSD bond | 0.007 | 0.006 |
| RMSD angles | 0.722 | 0.570 |
| Model-to-map fit, CC Mask | 0.77 | 0.77 |
| **Validation** |  |  |
| MolProbity | 1.85 | 1.42 |
| Clashscore | 6.86 | 4.15 |
| EMRinger score | 2.77 | 4.70 |
| Poor rotamers (%) | 0.00 | 0.45 |
| **Ramachandran** |  |  |
| Favored (%) | 92.46 | 96.59 |
| Allowed (%) | 7.22 | 3.31 |
| Outliers (%) | 0.32 | 0.10 |

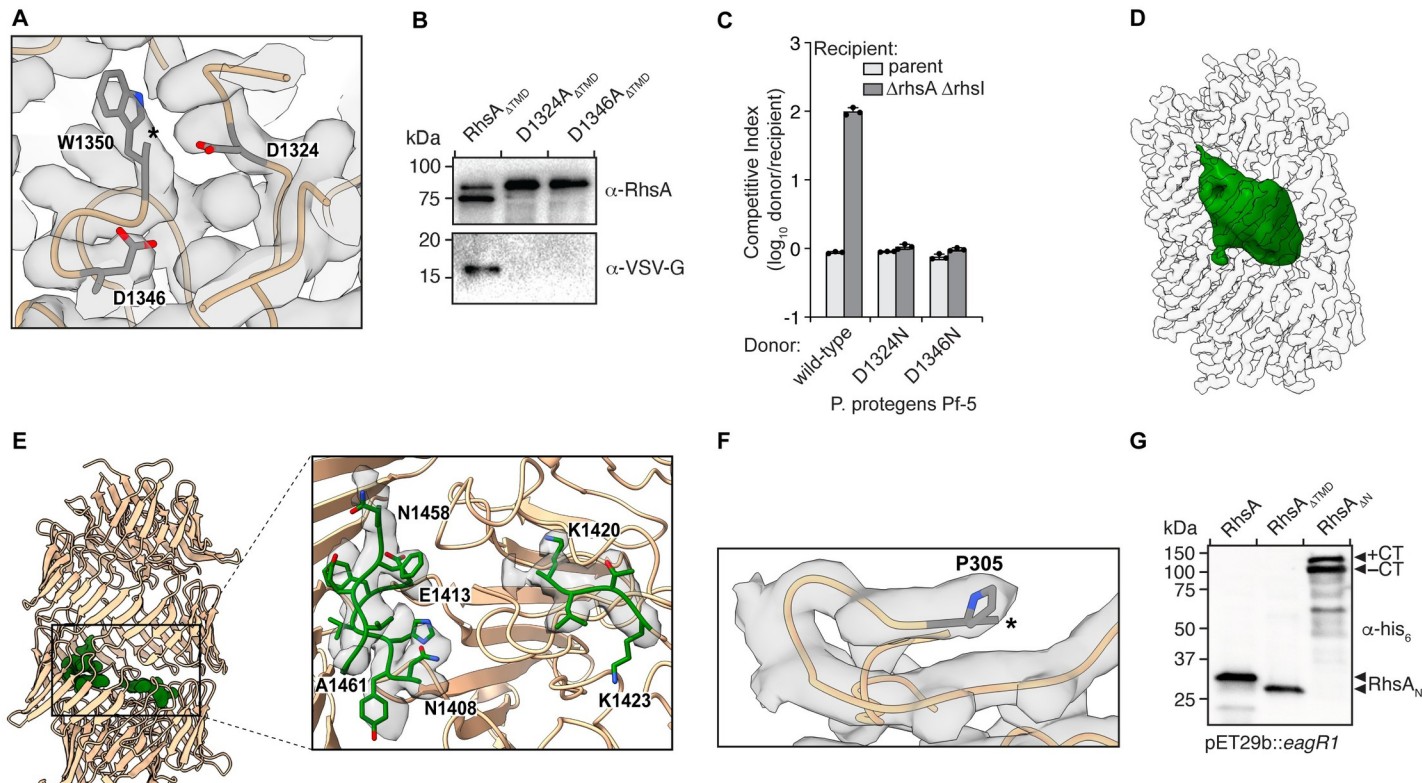

**Fig 2. Autoproteolysis of RhsA occurs at its N- and C-terminus.** (A) RhsA is autoproteolytically cleaved at its C-terminus at position W1350. The end of the connected density is indicated with an asterisk. Catalytic aspartates D1324 and D1346, are shown in stick representation. (B) Western blot analysis shows proteolytic cleavage of the C-terminal toxin domain. Mutation of either D1324 or D1346 to asparagine prevents autoproteolytic cleavage of the C-terminal RhsA toxin domain. Blots were performed against the Rhs barrel (α-RhsA) and against a C-terminal VSV-G epitope tag (α-VSV-G). (C) Outcome of intraspecific growth competition assays between the indicated *P. protegens* donor and recipient strains. Donor strains were competed against recipient strains that either contain (light grey) or lack the *rhsA-rhsI* effector-immunity pair (dark grey). The recipient strains also lack *pppA* to stimulate type VI secretion in donors [14]. Data are mean ± s.d. for *n* = 3 biological replicates and are representative of two independent experiments. (D) The toxin domain of RhsA is encapsulated by its cocoon-shaped Rhs repeat-containing domain. Difference map of the encapsulated toxin at extremely low-density threshold and low pass-filtered to 20 Å (green) is shown. The Rhs cocoon is depicted using a transparent space-filling representation at normal map threshold. (E) Regions of the toxin domain are stabilized inside the cocoon through interactions with the C-terminal autoprotease domain. The densities appear at the same density threshold as the rest of the map. The atomic models are shown in stick representation. (F) RhsA undergoes N-terminal cleavage at proline 305. The end of the density is indicated with an asterisk. (G) N-terminal cleavage occurs in RhsA and a mutant lacking the prePAAR-TMD (RhsA$_{\Delta TMD}$, residues 74-CT) but not in a mutant lacking the entire N-terminal region (RhsA$_{\Delta N}$, residues 297-CT). The indicated RhsA constructs were purified from *E. coli* and subject to Western blot and detected using an N-terminal His$_6$-tag antibody (α-his$_6$).

toxin domain (residues 1351–1486), indicating that these regions likely exhibit a high degree of flexibility.

The two molecules in the RhsA$_{\Delta TMD}$ dimer interact along their longitudinal axes and are slightly tilted with respect to one another. The dimer interface is comprised of several complementary hydrophilic surfaces indicating that it is mostly electrostatic in nature (S2A and S2B Fig). The 76 antiparallel β-strands of RhsA$_{\Delta TMD}$ spiral in an anticlockwise manner resulting in a large hollow cocoon-shaped structure with outer dimensions of 86 x 65 Å. The overall structure of the Rhs cocoon resembles that of other YD-repeat containing proteins, such as the BC component of Tc toxins [26,27] and human teneurin2 [28]. We also identified the conserved catalytic center of an aspartyl autoprotease that was first identified in Tc toxins [26,27]. In line with functioning to self-cleave the C-terminal toxin domain, we observed that the density of the cryo-EM map stops abruptly after tryptophan 1350 (Fig 2A). This position is in agreement with cleavage sites in other Rhs-related toxins that share the same PxxxxDPxG W/L/F

consensus sequence found in RhsA (S3A Fig), indicating that the C-terminal toxin domain is likely autoproteolytically cleaved similar to the HVR found in Tc toxins. To experimentally test the proteolytic activity of this motif, we mutated either aspartate 1324 or aspartate 1346, which are the RhsA residues in the equivalent position to the catalytic dyad of Tc toxin aspartyl autoproteases, to asparagine. Consistent with their proposed role in autocleavage and toxin release, these RhsA variants did not undergo autoproteolysis at their C-terminus (Fig 2B) and were substantially less toxic when overexpressed in *E. coli* compared to the wild-type protein (S3B Fig). We next introduced these mutations into the chromosome of *P. protegens* to test whether C-terminal autoproteolytic cleavage is indeed required for interbacterial competition. In contrast to a strain expressing wild-type RhsA, strains expressing the D1324N or D1346N variants of the protein were unable to outcompete RhsA-sensitive recipient bacteria indicating that C-terminal autocleavage is required for T6SS-dependent killing by this effector (Fig 2C). These findings mirror what has been observed for T6SS-exported Rhs proteins in *Aeromonas dhakensis* and *Enterobacter cloacae* suggesting that toxin domain liberation is a universal property of Rhs effectors [18,34].

We also identified additional density inside the Rhs cocoon corresponding to the C-terminal toxin domain, which is predicted to function as a DNase based on its sequence similarity to several characterized endonucleases and homology to other characterized Rhs effectors [35]. The density filling the cocoon is, for the most part, not well defined and is apparent only at a much lower density threshold compared to the rest of the map (Fig 2D), Nevertheless, we could build three β-strands of this domain, comprising residues 1408–1413, 1420–1423 and 1458–1461, respectively, all of which interact with the inner surface of the cocoon structure (Fig 2E). The interface between the β-strands and the cocoon is stabilized by both hydrophilic and hydrophobic interactions (Figs 2E and S4). In sum, these structural data indicate that most of the toxin domain is either only partially folded or very flexible.

Interestingly, the density inside the Rhs cocoon not only ends after tryptophan 1350 but also before proline 305 (Fig 2F), suggesting that RhsA is proteolytically cleaved at this position as well. However, the N-terminal region is presumably still associated with the cocoon structure because the protein was purified by affinity chromatography using an N-terminal His-tag. Therefore, we heat-denatured RhsA$_{WT}$ and RhsA$_{\Delta TMD}$ prior to SDS-PAGE to determine if the N-terminal domain was indeed cleaved. Consistent with our structural data, we observed bands that migrate at molecular weights consistent with the loss of ~304 and ~230 amino acids, respectively, corresponding to the 304 N-terminal residues of RhsA$_{WT}$ and 230 N-terminal residues of RhsA$_{\Delta TMD}$ (Fig 2G). To unambiguously determine the position of the cleavage site we performed *de novo* protein sequencing via LC-MS/MS. We found that the N-terminal peptide of the Rhs-cocoon detected by mass spectrometry exactly matches the proposed cleavage site at proline 305 (S5A Fig). In addition, we further validated cleavage at this position by generating a non-cleavable version of RhsA in which both residues H304 and P305 were mutated to alanine. This RhsA variant was unable to undergo autoproteolysis at its N-terminus (S5B Fig). Collectively, these data demonstrate that the protein is indeed cleaved between residues 304 and 305. By contrast, a truncation of RhsA lacking this entire N-terminal region, RhsA$_{\Delta N}$, did not undergo N-terminal proteolysis (Fig 2G). A similar observation was recently made for the T6SS effector TseI from *A. dhakensis*, which is also N-terminally cleaved even though it differs from RhsA at its N-terminus in that it lacks prePAAR, PAAR and a TMD [18]. This cleavage event was shown to be essential for the activity of TseI after its secretion by the T6SS. Two glutamate residues at the +7 and +8 position relative to the N-terminal cleavage site are responsible for the autoproteolysis of this effector [18]. However, these glutamates are not conserved in RhsA and are replaced by alanine (A312) and lysine (K313), which cannot act as catalytic center (S5C–S5E Fig). Therefore, we examined the direct vicinity of P305 in our

structure and found a cysteine residue (C538) flanked by two histidines (H530 and H555) that protrude from the wall of the Rhs barrel and are located near proline 305 (S5F Fig). Unlike the previously described glutamate residues, this site is conserved in class I prePAAR T6SS Rhs effectors, but does not exist in Tc toxins (S5G Fig) [18]. In addition, we identified three consecutive aspartates (D318-D320) in close proximity to the cleavage site.

To test the hypothesis of whether cysteine 538 or the other proximal residues are involved in N-terminal cleavage, we mutated these residues individually and performed a Western blot analysis against the N-terminus of the overexpressed constructs (S5H Fig). Contrary to our assumption, we found that mutation of C538 to alanine did not abolish the N-terminal cleavage. Mutations of other conserved residues near the cleavage site had also little effect. Only the triple mutant (D318-D320 to alanine) led to a minimal reduction of N-terminal cleavage. However, we believe that this could be an indirect effect induced by the mutationally altered conformation in this region similar to the construct where we mutated H530 to alanine.

Since there are no other obvious potential catalytically active residues besides the tested ones, we speculated that the encapsulated C-terminal toxin domain could be involved in catalyzing the N-terminal cleavage reaction. To test this, we generated a truncated form of RhsA lacking its C-terminal toxin domain (RhsA$_{\Delta tox}$) and examined N-terminal cleavage via Western blot. This analysis clearly showed that RhsA$_{\Delta tox}$ is still able to undergo autoproteolysis at the N-terminus (S5I Fig). Therefore, the N-terminal cleavage must occur via an as yet unknown mechanism.

## RhsA possesses a unique plug domain at its N-terminus

Our structural data show that the C-terminal aspartyl protease domain of RhsA seals the cocoon-shaped structure at one end, while the other end is capped by an N-terminal domain that adopts a smaller structure. This N-terminal domain of the barrel, formed by residues 268–386, caps the cocoon structure in a manner that is reminiscent of how a cork is used to plug a champagne bottle and thus we refer to it as the N-terminal plug domain (RhsA$_{plug}$) (Fig 3A–3E). The interface between the plug domain and the Rhs repeats is mainly stabilized by hydrophobic interactions (Fig 3A–3C) and a few hydrophilic interactions (S6A Fig).

In addition to the cork structure (residues 305–386), the plug domain contains an anchor helix (residues 289–302) and an N-terminal seal peptide (residues 268–275). The seal peptide fills a small opening in the cork as it leaves the cocoon and connects the plug domain to the unmodelled PAAR domain (Figs 3D–3E and S7A and S7B). In doing so, the seal peptide, together with the cork, closes off the cocoon entirely (S7B Fig). The anchor helix is amphipathic and stabilized by interactions with a hydrophobic patch of the Rhs repeats in this region (Figs 3E and 3F and S7C–S7E). The position of the anchor helix in our structure suggests its function is twofold. On the one hand it holds the N-terminal cleavage site in place. On the other hand, it ensures that the N-terminal plug domain remains stably attached to the cocoon, so that it remains sealed even after N-terminal cleavage.

The plug domain of RhsA possesses sequence and structural similarity to a domain found in the BC components of TcdB2-TccC3 [26] and YenBC [27] (Figs 3G and S8A–S8C) (18% sequence identity to TcdB2; 22% sequence identity to YenB). Interestingly, in Tc toxins this domain does not act as a plug that prevents the release of the toxin, but instead forms a negatively charged constriction through which the toxin domain is threaded prior to its translocation into a target cell [32]. A plug domain has also been described in teneurins (called fibronectin-plug, FN-plug) [28,29], however, it does not bear sequence or structural similarity to the plug domain of RhsA described herein. The FN-plug is narrower than the plug domains

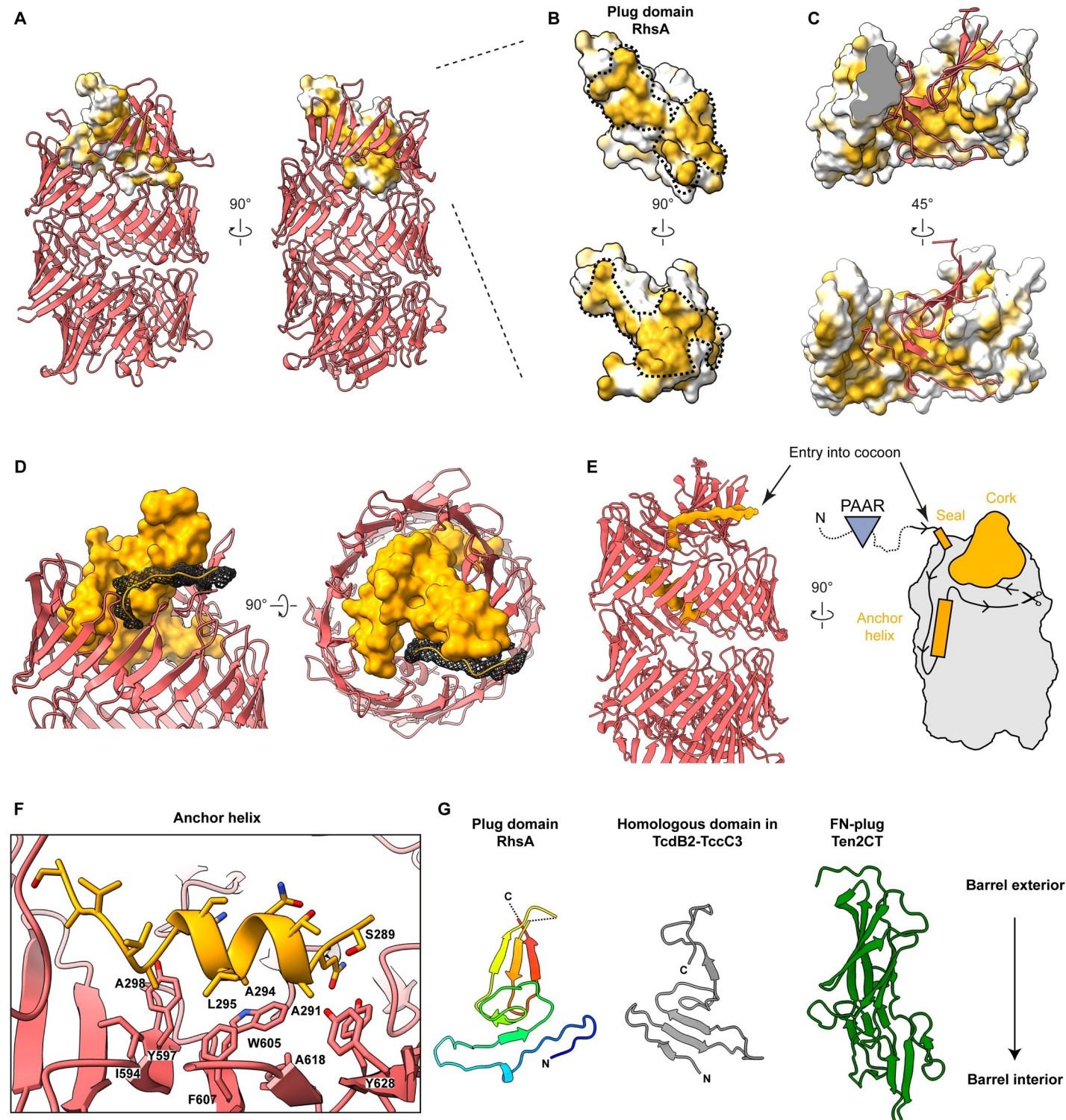

**Fig 3. A unique plug domain seals the Rhs barrel of RhsA.** (A) Surface representation of the cork domain of RhsA. The molecular surface is colored according to hydrophobicity where ochre and white indicate hydrophobic and hydrophilic regions, respectively. The Rhs barrel is shown as a cartoon. (B) Enlarged view of the cork domain of RhsA. The hydrophobic surface spirals around the domain as indicated by the black dashed line. (C) The upper Rhs repeats of RhsA possess complementary hydrophobic patches to those found on its plug domain (shown in cartoon representation, red). (D) The cocoon structure of RhsA is closed off by an N-terminal plug comprised of a 'cork' domain (orange, density representation), a 'seal' peptide (mesh), and an anchor helix (E, F). The seal and the cork density together form a cap structure. (E) The seal peptide, which also functions as the linker to the unmodelled PAAR domain, not only complements the shape of the cork but is also the entry

point of the N-terminal part of the protein into the inside of the cocoon. The cocoon remains stably bound to the cleaved N-terminal region, including the PAAR domain, due to the anchor helix inside the cocoon. (F) The anchor helix of RhsA is stabilized by hydrophobic interactions with the inner wall of the Rhs repeats. (G) Cartoon representation of the cork region of the RhsA plug domain and a comparison with the homologous N-terminal plug domain of *Photorhabdus luminescens* TcdB2 (grey, PDB ID: 6H6G) and the unique FN-plug domain of human teneurin2 (green, PDB ID: 6FB3). The lower part of each depicted plug domain inserts into each of their respective Rhs barrels. The cork domain of RhsA is colored in rainbow.

of RhsA and Tc toxins and extends further into the Rhs cocoon structure (Fig 3G) making numerous hydrophilic interactions with residues lining the inside of the YD-shell [28].

Our structure suggests that in contrast to Tc toxins the plug domain of RhsA tightly seals its Rhs cocoon. Because the plug domain appears to strongly interact with the Rhs barrel, mechanical removal of this entire domain after translocation of the cocoon into the target cell cytosol seems unlikely. Instead, we propose that only the N-terminal peptide seal of the RhsA plug is pulled out during T6SS-dependent delivery of RhsA into a susceptible bacterial cell. The seal peptide is part of the N-terminal region that is proteolytically cleaved, so we speculate that it would be more easily removed compared to the entire plug domain, which is held in place by the anchor helix that exists downstream of the N-terminal cleavage site. Penetration of the outer membrane and peptidoglycan layer as well as translocation of RhsA through the target cell inner membrane could provide the mechanical force to remove the seal peptide, creating a channel through which the unfolded toxin domain could exit. Since the plug domain resembles the constriction site in Tc toxins that effectors must pass during the initial translocation process, the same process is conceivable for the release of the C-terminal toxin domain of RhsA.

## Comparison between Rhs proteins of known structure

A common feature of all Rhs proteins that have been structurally characterized to date is the cocoon structure formed by the Rhs repeats. While the Rhs cocoons from RhsA and teneurin2 are comprised of three β-helical turns of Rhs repeats, the cocoons of Tc toxins have four turns and therefore have bigger overall dimensions and an internal cavity with larger volume. Consistent with this observation, the effector domain inside Tc toxin cocoons is larger (~30 kDa) than the toxin domain of RhsA (~15 kDa). All characterized Rhs proteins contain a conserved C-terminal region. For Tc toxins and RhsA this domain functions as aspartyl autoprotease whereas in teneurins it acts as a YD-shell plug that is not cleaved (Fig 4 blue). In all instances, the C-terminal conserved region comprises 14 Rhs repeats that spiral into the inside of the cocoon and seal it at one end.

The plug domains that close off the N-terminal end of the cocoon structures are more variable than their C-terminal counterparts (Fig 4 orange). While the barrel of teneurins is sealed with an FN-plug that is mainly held in place by hydrophilic interactions with the interior of the barrel, TcdB2 from *Photorhabdus luminescens* and other BC components of Tc toxins close the cocoon using a β-propeller domain that acts as gatekeeper for toxin release. RhsA uses an N-terminal plug domain, which is homologous to the domain that forms the constriction site in Tc toxins. Both TcBC and RhsA encapsule a toxic effector, whereas teneurins act as scaffolding proteins with an empty cocoon (Fig 4 red). Nevertheless, teneurins encode a C-terminal ancient toxin component that sits outside of the barrel and is inactive (ABD Tox-GHH, Fig 4 red). While RhsA contains two autoprotease sites, namely an aspartyl protease that is responsible for the cleavage of the C-terminal toxic effector and an unknown protease site that cleaves the N-terminal 304 residues of the protein, Tc toxins only contain the former site and teneurins none at all.

In conclusion, based on its unique plug domain and its fusion to an N-terminal PAAR domain, we propose that our RhsA structure represents the founding member of a third structural class of Rhs repeat containing proteins (Fig 4).

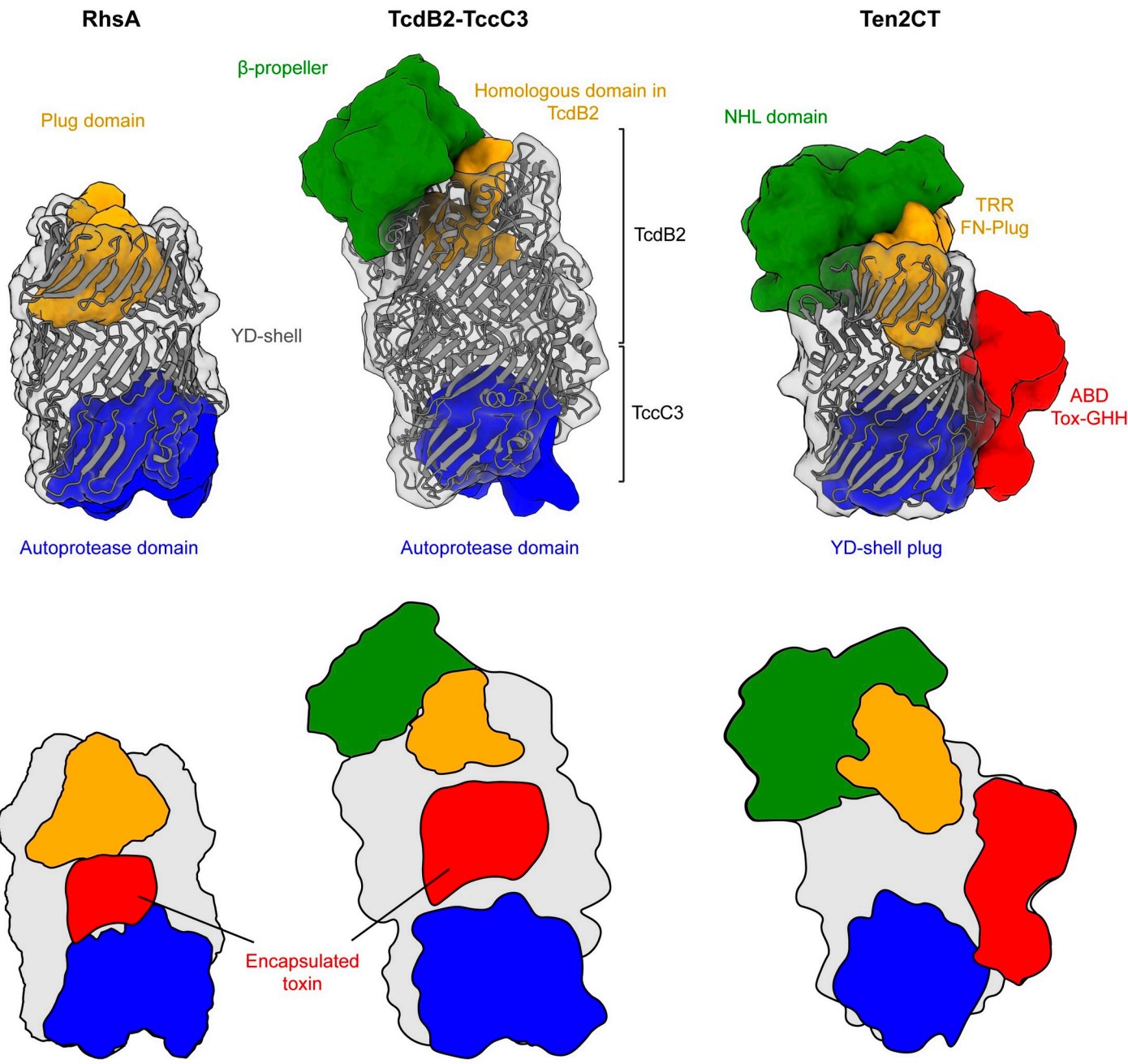

**Fig 4. Structural comparison of Rhs repeat containing proteins.** The C-terminal autoproteolysis domain (left and middle) or YD-shell plug (right) is structurally conserved among diverse Rhs proteins (blue). RhsA and BC components of Tc toxin complexes encapsulate their toxic effector domains (red) whereas in teneurin proteins the toxin domain is appended to the outside of the Rhs barrel (red). A distinguishing feature of these Rhs proteins is the unique N-terminal plug domain for each protein family (orange). RhsA is capped by a cork-like plug domain that seals the Rhs barrel (orange). In Tc toxins, the homologous plug domain acts as constriction site and the cocoon is sealed off by a β-propeller domain (green). Teneurin proteins are capped with a non-homologous FN-plug (orange) that is stabilized by an NHL domain (green). The lower row shows schematic representations of the domain organizations.

## Architecture of the pre-firing complex

To investigate how RhsA is mounted onto VgrG1 we set out to determine the structure of the secretion competent pre-firing complex (PFC) comprising VgrG1 in complex with full-length RhsA and EagR1. We purified the complex as described previously [14] and examined it by

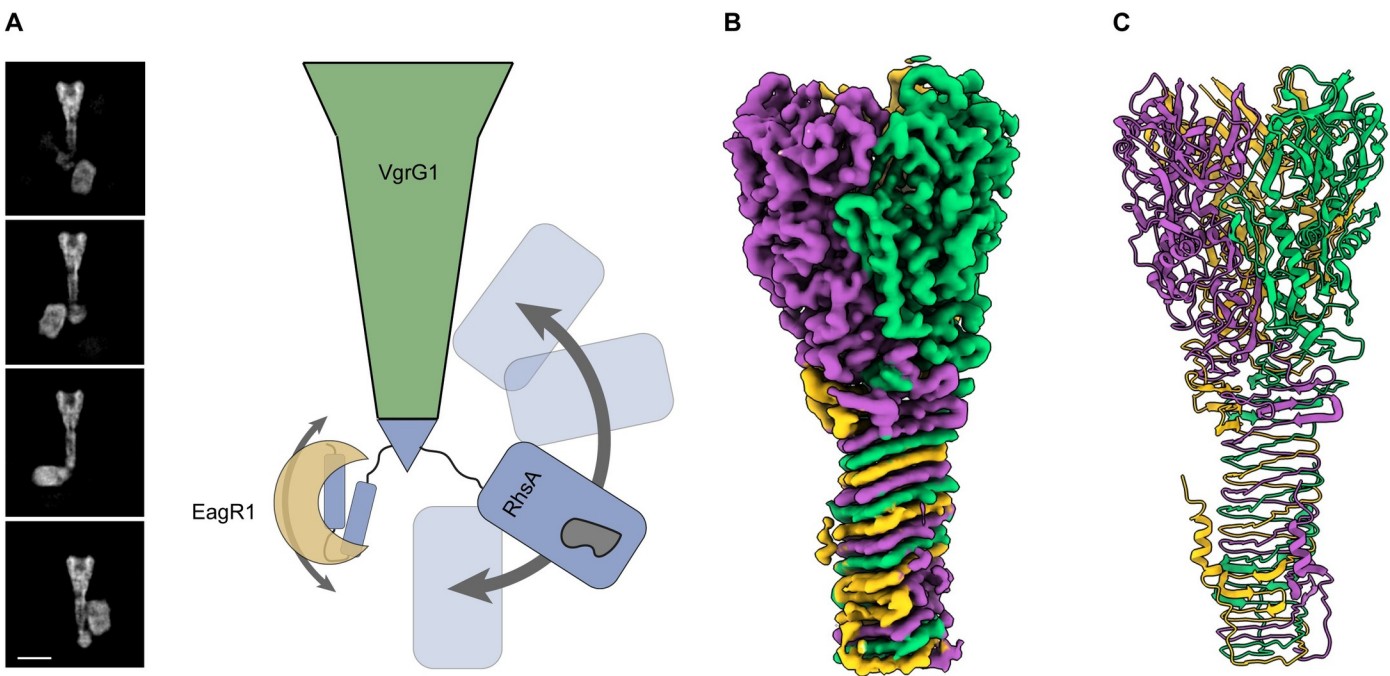

**Fig 5. High-resolution structure of *P. protegens* VgrG1.** (A) The RhsA barrel of the assembled pre-firing complex (PFC) displays high positional flexibility relative to VgrG1. Representative cryo-EM 2-D class averages depicting flexibility are shown. Scale bar, 10 nm. (B) Cryo-EM density and (C) ribbon representation of the molecular model of the *P. protegens* VgrG1 trimer viewed along the long axis of the protein. Each protomer is colored differently to highlight their positions within the homotrimeric VgrG1 spike.

single particle cryo-EM. The 2D class averages enabled us to unequivocally characterize the arrangement of the subunits within the complex (Fig 5A). Expectedly, both EagR1 and RhsA are located at the tip of VgrG1. Interestingly, a large fraction of PFCs contained RhsA dimers instead of monomers (S9B Fig), similar to what was observed in our analysis of RhsA$_{\Delta TMD}$ alone. This demonstrates that the loading of two RhsA molecules onto one VgrG1 is sterically possible although it is unlikely to occur *in vivo* given that current data indicates a single VgrG homotrimer caps the Hcp tube [36].

As is the case for the previously characterized VgrG1-EagT6-Tse6 complex from *P. aeruginosa* [16], T6SS effectors can adopt multiple positions relative to their cognate VgrG spike protein. Unfortunately, this conformational heterogeneity prevented us from obtaining a high-resolution 3D reconstruction of the entire complex (S1 Movie). Instead, we applied $C_3$ symmetry during processing to determine the three-dimensional structure of *P. protegens* VgrG1 (Fig 5B). In applying this symmetry operator, the RhsA and EagR1 components of the complex, which do not adopt this symmetry, are averaged out during image processing. The cryo-EM map of VgrG1 reached a resolution of 3.3 Å and allowed us to build almost the complete atomic model of the protein comprising residues 8–643 (Fig 5C and Table 1). As expected, given its high sequence homology, the structure of VgrG1 from *P. protegens* is nearly identical (71% sequence identity, 81% sequence similarity) to our previously determined VgrG1 structure from *P. aeruginosa* (r.m.s.d of 1.035 between 544 pruned Cα atoms; r.m.s.d of 1.428 across the complete structure, S9B and S9C Fig) [16] even though their respective effectors, RhsA and Tse6, bear no sequence or structural similarity to one another beyond their N-terminal pre-PAAR and PAAR domains. Intriguingly, we identified two spherical densities in the center of the β-sheet prism of the VgrG1 trimer (S9E Fig). Since we also observed these densities previously in the VgrG1 structure from *P. aeruginosa* [16], we speculate that this may be a common

feature of VgrG1 proteins. Based on the exclusive clustering of positively charged residues around this density and its overall size, we hypothesize that it corresponds to either a phosphate or sulfate anion, which probably helps to stabilize the core of VgrG1 given that in the absence of this anion, the presence of positively charged residues in the core of the protein would be energetically unfavorable.

## Model of RhsA firing events and toxin release

Based on the collective structural and functional data presented in this work, combined with the findings of other recently published [18,34] work on T6SS-exported Rhs proteins, we propose a model for cytoplasmic delivery of RhsA and suggest a possible release mechanism for the toxin domain of the effector (Fig 6).

First, RhsA is expressed and autoproteolytically cleaved at defined N- and C-terminal positions. The N-terminal domain comprising the prePAAR motif, TMD region, PAAR domain and the linker to the Rhs barrel likely all simultaneously interact with the EagR1 chaperone and this complex remains associated with the RhsA cocoon. The C-terminal domain comprising the detached toxin domain is partially unfolded and remains inside the cocoon.

EagR1 then shields the transmembrane domains of RhsA from the aqueous milieu as the effector is loaded onto VgrG1 in the cytoplasm of the T6SS-containing cell resulting in a mature PFC. The exact location of T6SS effector delivery in the target cell remains unclear and may differ depending on the T6SS tail spike complex being exported [37,38]. Nonetheless, most characterized PAAR effectors exert their catalytic activity in the cytosol of the target cell, i.e. (p)ppApp synthetases, ADP-ribosyl transferases, DNases and $NAD^+/NADP^+$ hydrolases [11,35,39,40]. We propose that the RhsA-loaded VgrG1 tip is delivered to the periplasm as we have previously suggested for the Tse6 effector and that the PFC spontaneously enters the target cell inner membrane [16]. It remains unclear at which step the chaperone is stripped off; however, Coulthurst and colleagues detected a secreted T6SS Rhs effector in *Serratia* by mass spectrometry and did not detect its cognate Eag chaperone [41]. This finding supports a model whereby EagR1 dissociates from the spike complex during the loading event of the PFC onto an Hcp tube inside the lumen of the T6SS baseplate. But given the observation that the TMD helices are unstable without their protective chaperones [14], it is also conceivable that EagR1 is removed shortly before insertion into the membrane of the target cell. As proposed previously for Tse6, we suggest that the transmembrane helices of the TMD region spontaneously enter the inner membrane of the target cell [16].

Given the rigid structure of RhsA, we propose that the barrel remains intact during the firing and translocation events. Two different scenarios are conceivable for the release of the toxin domain from the cocoon and its delivery into the cytosol of the target cell if the VgrG spike only protrudes into the periplasm. In one scenario, translocation of the linker between the PAAR domain and the plug domain would lead to the removal of the seal peptide. This would result in the opening of a passageway through which the toxin domain could be threaded in an unfolded or partially unfolded state and subsequently translocated, with the assistance of the N-terminal transmembrane helices, into the cytosol. In this case, the RhsA barrel would remain in the periplasm but interact directly with the inner membrane of the target cell. This model has similarity to the currently proposed model for translocation of diphtheria toxin from *Corynebacterium diphtheriae*. In this case, two α-helical hairpins insert into the endosomal membrane and form a translocation channel through which the toxin is delivered into the cytosol of the intoxicated cell [42]. The refolding of diphtheria toxin in the target cell cytosol is thought to drive the translocation through the loosened-up membrane structure created by the two inserted transmembrane helices. Moreover, it was previously shown [18]

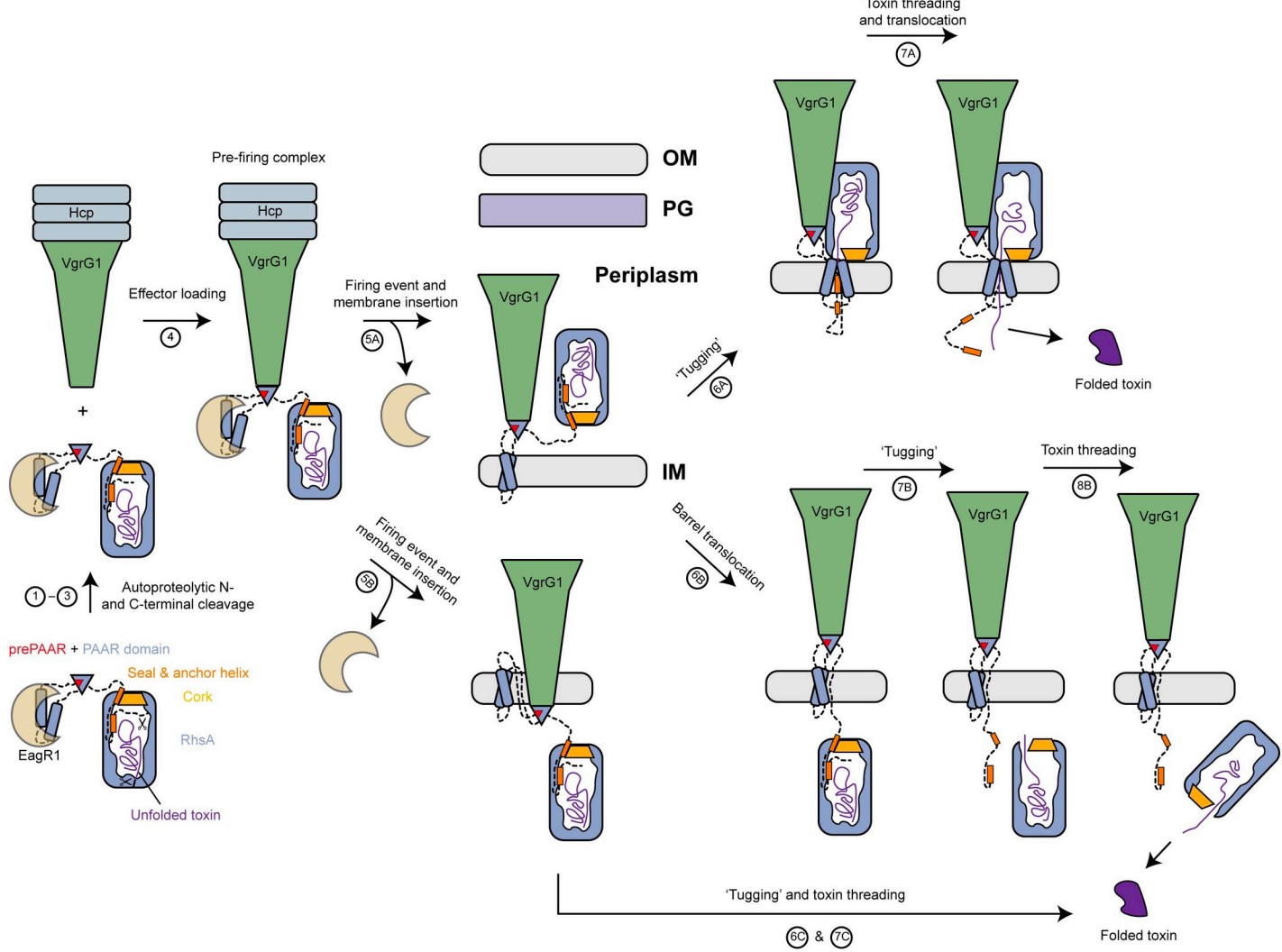

**Fig 6. Model of T6SS-dependent delivery of RhsA into the cytoplasm of a susceptible bacterial cell.** (1–3) RhsA undergoes N- and C-terminal autoproteolytic processing. The prePAAR motif 'completes' the PAAR domain fold. The EagR1 chaperone solubilizes the two transmembrane helices of RhsA to facilitate loading onto its cognate cytoplasmic VgrG1 (via the prePAAR + PAAR domain). (4) The secretion competent RhsA effector is loaded onto the VgrG1 spike. (5) During a firing event the EagR1 chaperone is dissociated from the complex and the T6SS injects the PFC into target cells where it crosses the peptidoglycan (PG) layer and inserts into the inner membrane (IM) (5A). Alternatively, the tip of the VgrG1 spike could be directly delivered into the cytosol along with RhsA (5B). (6–8) Two different scenarios (6A or 6B,C) are proposed as possible mechanisms for toxin domain release from the cocoon into the cytosol of the target cell. Either the toxin domain alone (mechanism 6A) or the entire RhsA barrel is translocated across the inner membrane (mechanism 6B). In both cases, the seal of the cocoon is likely removed by translocation-induced pulling and the energy required for release and translocation of the toxin domain out of the Rhs cage is probably driven by its spontaneous refolding in the prey bacterium's cytosol.

that the two VgrG-interacting domains of the T6SS effector TseI from *A. dhakensis*, denoted as VIRN and VIRC, which are equivalent to the cleaved N- and C-terminal domains of RhsA, interact directly with one other even in the absence of the Rhs shell. Hence, it seems plausible that once the seal peptide is removed from RhsA, the unfolded C-terminal toxin domain will be threaded out of the barrel via interactions with its N-terminal domain.

In a second scenario, the TMD region would facilitate the translocation of the entire RhsA protein. Mechanical forces during translocation in combination with hydrophobic interactions in the membrane could lead to 'tugging' of the N-terminal domain and a release of the toxin domain from the cocoon. However, since the barrel is large (86 x 65 Å), its transport

across the membrane is relatively unlikely. In both scenarios, the energy required for the release and translocation of the toxin domain into the target cell cytosol would be driven by spontaneous toxin refolding [43].

An alternative route would be the direct delivery of the barrel into the cytosol. This would require that the PFC not only penetrates the outer membrane, but also the inner membrane to reach the cytosol of the target cell. In this case, the cocoon would be located in the cytosol of the target cell, and the toxin domain would be released analogously to above-described scenario B. However, it is unclear what the role of the TMD would be in this alternative delivery mechanism.

Besides the proposed scenarios, other pathways for RhsA delivery and toxin release that have not been described are also conceivable. For example, periplasmic or inner membrane proteins found in the target cell could be involved in the active transport of the toxin from the periplasm into the cytosol.

Overall, this study provides detailed molecular insights into the autoproteolytic processing of Rhs effectors and its importance for toxin release inside the target cell. It not only enhances our knowledge about Rhs effector function, but also lays a foundation for a mechanistic understanding of how the T6SS machinery functions. The unique ability of the T6SS to mediate contact-dependent killing of a wide range of bacteria may enable development of novel therapeutics for selective depletion of drug-resistant bacterial pathogens.

## Materials and methods

### Bacterial strains and culture conditions

*Pseudomonas protegens* Pf-5 (S1 Table) was grown in Lysogeny Broth (LB) (10 g L$^{-1}$ NaCl, 10 g L$^{-1}$ tryptone, and 5 g L$^{-1}$ yeast extract) at 30˚C or on solid LB containing 1.5% or 3% agar. Media were supplemented with gentamicin (30 μg mL$^{-1}$) and irgasan (25 μg mL$^{-1}$) as needed.

*Escherichia coli* strains XL-1 Blue, SM10 and CodonPlus (DE3) were used for plasmid maintenance and toxicity experiments, conjugative transfer and protein overexpression, respectively (S1 Table). All *E. coli* strains were grown at 37˚C in LB medium. Unless otherwise noted, media was supplemented with 150 μg mL$^{-1}$ carbenicillin, 50 μg mL$^{-1}$ kanamycin, 200 μg mL$^{-1}$ trimethoprim, 15 μg mL$^{-1}$ gentamicin, 0.25–1.0 mM isopropyl β-D-1-thiogalactopyranoside (IPTG), 0.1% (w/v) rhamnose or 40 μg mL$^{-1}$ X-gal.

### DNA manipulation and plasmid construction

All primers used in this study were synthesized by Integrated DNA Technologies (IDT). Molecular biology reagents (Phusion polymerase, restriction enzymes and T4 DNA ligase) were obtained from New England Biolabs (NEB). Sanger sequencing was performed by Genewiz Incorporated.

Heterologous expression plasmids: pETDuet-1, pET29b and pSCrhaB2-CV. Splicing by overlap-extension PCR was used to make mutant constructs. Standard restriction enzyme-based cloning procedures were subsequently used to ligate wild-type or mutant PCR products into the plasmid of interest.

### Generation of *P. protegens* mutants

In-frame chromosomal deletion mutants in *P. protegens* were made using the pEXG2 plasmid as described previously for *Pseudomonas aeruginosa* [44]. Briefly, ~500 bp upstream and downstream of target gene were amplified by standard PCR and spliced together by overlap-extension PCR. The resulting DNA fragment was ligated into the pEXG2 allelic exchange

vector using standard cloning procedures (S2 Table). Deletion constructs were transformed into *E. coli* SM10 and subsequently introduced into *P. protegens* via conjugal transfer. Merodiploids were directly plated on LB (lacking NaCl) containing 5% (w/v) sucrose for *sacB*-based counter-selection. Deletions were confirmed by colony PCR in strains that were resistant to sucrose, but sensitive to gentamicin. Chromosomal point mutations or epitope tags were constructed similarly with the constructs harboring the mutation or tag cloned into pEXG2. Sucrose-resistant and gentamicin-sensitive colonies were confirmed to have the mutations of interest by Sanger sequencing of appropriate PCR amplicons.

### *Pseudomonas* growth competition assays

Recipient *P. protegens* strains contained a $\Delta pppA$ mutation to stimulate T6SS effector secretion and induce a 'counterattack' from *P. protegens* donor strains [45]. Recipient strains were also marked with a tetracycline-resistant, *lacZ*-expression cassette at a neutral phage site (*attB*) to differentiate from unlabeled donor strains.

Stationary-phase overnight cultures of *P. protegens* donors and recipients were mixed in a 1:1 (v/v) ratio and relative abundance of donor:recipient was determined by plating part of the competition mixture on LB plates with 40 µg mL-1 X-gal. Ten microlitres of each competition mixture was then spotted on a 0.45 µm nitrocellulose membrane that was overlaid on a 3% LB agar plate and incubated face up at 30˚C. Competition spots were harvested after 20–25 hours by resuspending in LB and counting CFU by plating on LB agar with 40 µg mL$^{-1}$ X-gal. The final ratio of donor:recipient colony forming units were then normalized to the initial ratios of donor and recipient strains and reported as the $\log_{10}$ of the competitive index.

### Toxicity assays

Wild-type RhsA and the various RhsA truncations and site-specific mutants used in this study were cloned into the rhamnose-inducible pSCrhaB2-CV vector [46]. RhsI was cloned into the IPTG-inducible vector pPSV39 [12]. The various RhsA expressing pSCrhaB2-CV plasmids were co-transformed into *E. coli* XL-1 Blue with pPSV39::*rhsI*. Stationary-phase overnight cultures containing these plasmids were serially diluted $10^{-6}$ in 10-fold increments and each dilution was spotted onto LB agar plates containing 0.1% (w/v) L-rhamnose, 500 µM IPTG, trimethoprim 250 µg mL$^{-1}$ and 15 µg mL$^{-1}$ gentamicin. Photographs were taken after overnight growth at 37˚C.

### Protein expression and purification

Wild-type RhsA, its N-terminal (C538A, D322A, H530A, D318-320A) and C-terminal autoproteolysis mutants (D1346A, D1364N) were cloned into MCS-1 of pETDuet-1 and co-expressed with RhsI, which was cloned into MCS2. Plasmids were co-transformed into *E. coli* BL21 Codon Plus alongside a pET29b vector expressing EagR1.

### Purification of RhsA$_{\Delta TMD}$ and VgrG1 for cryo-EM

RhsA$_{\Delta TMD}$ was co-expressed with RhsI using pETDuet-1 (see S2 Table for details). VgrG1 was expressed in isolation using pET29b. Plasmids were individually expressed in *E. coli* BL21 Codon Plus. Strains harboring pETDuet-1 expressing RhsA$_{\Delta TMD}$-RhsI or pET29b expressing VgrG1 were inoculated in separate flasks containing 100 mL LB with selection and incubated overnight in a shaking incubator at 37˚C. Following 14-18hr incubation, the culture was subinoculated (1/50 dilution) into four flasks each with 1 litre of LB and appropriate antibiotic selection. Cultures were initially incubated at 37˚C until the culture reached an OD of ~0.3.

The incubator was subsequently cooled to 18˚C and each culture induced with 1 mM IPTG upon reaching an OD of 0.6–0.7. Cultures were harvested by centrifugation at 9,800 *g* for 10 minutes. Pellets were resuspended in 30 mL lysis buffer (50 mM Tris-HCl, 300 mM NaCl, 10 mM imidazole) and lysed by sonication (6 x 30 second pulses, amplitude 30%) and then spun at 39,000 *g*. Cleared lysates were applied to a Ni-NTA gravity flow column equilibrated using lysis buffer. The column was washed with the lysis buffer three times and the samples were eluted in 3 mL of elution buffer (lysis buffer with 400 mM imidazole). The samples were applied to a HiLoad 16/600 Superdex 200 column equilibrated in 20 mM Tris-HCl pH 8.0 150 mM NaCl and collected in the same buffer. The sample was reinjected into a Superose 6 5/150 column using the same buffer for EM analysis and *de novo* protein sequencing.

## Western blot analyses

Western blot analyses were performed using rabbit anti-RhsA (diluted 1:5,000, Genscript, custom polyclonal [14]), rabbit anti-FLAG (diluted 1:5,000, Sigma), rabbit anti-VSV-G (diluted 1:5,000; Sigma) and mouse anti-His$_6$ (diluted 1:5,000, Genscript) and detected with anti-rabbit or anti-mouse horseradish peroxidase-conjugated secondary antibodies (diluted 1:5,000; Sigma). Development of western blots was completed using chemiluminescent substrate (Clarity Max, Bio-Rad) and imaged with the ChemiDoc Imaging System (Bio-Rad).

## *De novo* protein sequencing via LC-MS/MS

RhsA$_{\Delta TMD}$ was purified as described above. A concentrated sample was applied onto an SDS-PAGE. The band corresponding to the N- and C-terminally autoproteolyzed Rhs$_{cage}$ was excised from the gel and analysed by LC-MS/MS at the Proteome Factory AG company. The original data is provided as supplementary data (S1 Dataset).

## Negative stain electron microscopy

Four microliters of sample at a concentration of 0.005 mg/ml were applied to freshly glow-discharged carbon-coated copper grids. After 90s incubation time, excess sample was blotted away with Whatman No. 4, then washed twice with four microliters purification buffer and once with 0.75% (w/v) uranyl formate. A second batch of staining solution was incubated on the grid for 90s before excess was again blotted away. Grids were air-dried and imaged on a JEOL JEM-1400 microscope, equipped with a LaB$_6$ cathode and 4k × 4k CMOS detector F416 (TVIPS), operating at 120 kV.

## Sample vitrification and data collection

Three microliters of the EagR1-RhsA-VgrG1 complex, at a concentration of 0.1 mg/ml, were applied to a freshly glow-discharged holey carbon grid (QF 2/1 200-mesh). The grid was blotted for 3s (blot force -5, drain time 0.5 s, 8˚C, 100% humidity) and immediately plunged into nitrogen cooled liquid ethane using a Vitrobot Mark IV (Thermo Fisher Scientific). Data collection was performed on a C$_S$-corrected Titan Krios (Thermo Fisher Scientific) operating at 300 kV in an automated fashion using EPU (Thermo Fisher Scientific). Movies were recorded on a Falcon 3 detector in linear mode at a nominal magnification of 59,000x with a calibrated pixel size of 1.1 Å/pixel. Image stacks were acquired in a defocus range from -1.2 to -2.2 μm with an accumulated dose of 90 e$^-$/Å$^2$ fractionated over 40 frames with a total exposure time of 1.5 s.

For RhsA$_{\Delta TMD}$, three microliters of sample, at a concentration of 4 mg/ml, were applied to a freshly glow-discharged holey carbon grid (QF 1.2/1.3 200-mesh). The grid was blotted for

3s (blot force -5, drain time 0.5 s, 8˚C, 100% humidity) and immediately plunged into liquid ethane using a Vitrobot Mark IV (Thermo Fisher Scientific). The grid was transferred to a Titan Krios (Thermo Fisher Scientific) operating at 300 kV equipped with a GIF BioQuantum energy filter (Gatan), set to a slit width of 20 eV, and K3 Summit Detector (Gatan). Movies were recorded in counting mode at a nominal magnification of 105,000x with a calibrated pixel size of 0.91 Å/pixel in an automated fashion using EPU (Thermo Fisher Scientific). Image stacks were acquired in a defocus range from -0.8 to -2.2 μm with an accumulated dose of 61 e⁻/Å² fractionated over 20 frames with a total exposure time of 2 s.

Both datasets were monitored live with TranSPHIRE [47] to evaluate i.e. the defocus range and astigmatism. Pre-processing was performed on-the-fly in TranSPHIRE including drift-correction and dose-weighting with MotionCor2 [48], CTF estimation on dose-weighted micrographs with CTFFIND4 [49] and picking using the general model of crYOLO [50].

## Cryo-EM image processing

After preprocessing in TranSPHIRE all processing steps were carried out in the SPHIRE [51] software package unless otherwise stated. Images with a resolution limit less than 6 Å were unselected using the graphical CTF assessment tool in SPHIRE for both datasets.

In case of the PFC, particles were extracted with box size of 408 x 408 pixels from 1250 good micrographs. Reference-free 2D classification and cleaning of the dataset was performed with the iterative stable alignment and clustering approach ISAC [52] implemented in SPHIRE. ISAC was performed at pixel size of 6.29 Å/pix. Using the Beautifier tool, the original pixel size was restored creating sharpened 2D class averages showing high-resolution details. A subset of particles showing clear high-resolution details were selected for structure refinement. 3D refinement was performed in MERIDIEN imposing $C_3$ symmetry with a 25 Å lowpass-filtered reference of our previously determined VgrG1 structure (EMD-0136). The two half-maps were combined with the PostRefiner tool in SPHIRE using a soft mask and automatic estimation of B-factors. Details of the processing workflow are shown in S9 Fig.

For RhsA$_{\Delta TMD}$, particles were extracted from 13,090 good micrographs with a final box size of 288 x 288 pixels. Particles were subjected to ISAC which was performed at pixel size of 3.3 Å/pix. A subset of particles showing high-resolution features were selected from beautified class averages. An initial reference for refinement was generated from the beautified averages using RVPER. All refinements and classifications were performed with implied $C_2$ symmetry. Subsequent 3D refinement of the good particles with MERIDIEN yielded a map of overall 3.6 Å resolution but showed clear resolution anisotropy in the peripheries of the barrels. We imported the particle stack into RELION 3.1.0 [53] with projection parameters obtained from MERIDIEN. Particles were classified into 4 classes without image alignment (T = 4, $C_2$ symmetry, soft mask) using the map from MERIDIEN as reference low-pass filtered to 30 Å. The particles belonging to the class displaying the highest resolution were selected for another round of MERIDIEN after removal of duplicated particles (minimum inter-particle distance threshold 100 Å). Particles were further CTF-refined and polished in Relion. The final refinement was performed with MERIDIEN. The final map was evaluated using the 3D FSC tool [54]. Map for Fig 1 and was postprocessed with the DeepEMhancer [55] using the high-resolution model. Local resolution was estimated with a normal postprocessed map in SPHIRE. Details of the processing workflow are shown in S1 Fig.

## Model building, refinement and validation

The previously obtained VgrG1 structure from *P. aeruginosa* (PDB ID: 6H3L) was docked into the map as rigid body in UCSF Chimera [56]. The *P. protegens* VgrG1 sequence was manually

adjusted in Coot [57] and iteratively refined in Phenix [58] and ISOLDE [59]. Model validity was assessed in Phenix with MolProbity [60]. Final model statistics are given in Table 1.

For RhsA$_{\Delta TMD}$, initial backbone traces were identified with automated model building software Buccaneer [61]. The model was manually adjusted to completion in Coot [57] using both the DeepEMHanced and a normal postprocessed map to exclude any bias. Stabilized β-strands of the toxin domain were identified via secondary structure prediction and manually placed inside the densities. The single interpretable α-helix was not predicted by secondary structure predictions. Instead, we manually check the rest of the sequence to determine the sequence register. We placed residues 289–302 in this density guided by AlphaFold [62] predictions. This was moreover guided by the matched hydrophobicity towards the interacting Rhs repeats. To identify residues corresponding to the linker density which complements the plug domain, we searched in the vicinity of the helix sequence for residues encoding bulky side chains. F271 served as anchor point to build residues 268–275 guided by AlphaFold [62] predictions.

Iterative refinement of the model in Phenix [58] and ISOLDE [59] was performed until convergence. Model validity was assessed in Phenix with MolProbity [60]. Final model statistics are given in Table 1.

Figures were prepared in Chimera X [63]. Multiple sequence alignments and secondary structure predictions were calculated using the MPI Bioinformatics toolkit [64,65] and visualized for creation of figures in Jalview [66].

The cryo-EM maps of VgrG1 and RhsA have been deposited in the Electron Microscopy Data Bank (EMDB) under the accession codes of EMD-13843 and EMD-13867, respectively. The refined models for VgrG1 and RhsA were uploaded in the PDB and the entries have the IDs 7Q5P and 7Q97, respectively.

## Supporting information

**S1 Fig. Cryo-EM processing workflow used to obtain the structure of RhsA$_{\Delta TMD}$.** (A) Purification of RhsA$_{\Delta TMD}$ via size-exclusion chromatography using a Superose 6 5/150 increase column. Molecular weight standards are indicated at their respective elution volumes. The grey bar denotes pooled and concentrated fractions used for cryo-EM analysis. The same material was analyzed for purity via semi-denaturing SDS-PAGE imaged with a stain-free filter. (B) Representative cryo-EM micrograph of RhsA$_{\Delta TMD}$ used for structural determination. Scale bar EM micrograph, 100 nm. Particles were picked with the general model of crYOLO. The rest of the processing workflow is indicated and summarized in the Materials and Methods section. Scale bar 2D class averages, 10 nm. Used software packages are highlighted. Orange font depict steps carried out in Relion. The final map was calculated using MERIDIEN and postprocessed with DeepEMhancer. (C) Angular distribution plot of the final reconstruction. (D) Local resolution estimates visualized on a map postprocessed in SPHIRE. (E) Fourier shell correlation plot calculated from two independently processed maps. Resolution estimation is reported at the gold standard cutoff of 0.143. (F) Resolution anisotropy was assessed with the 3DFSC online server tool. (G) Selected regions of the map are shown as transparent surface with the built atomic models as stick representations.
(TIF)

**S2 Fig. Structural analysis of the dimer interface of RhsA.** (A) Close up view of the dimer interface of the two RhsA molecules at the central symmetry axis. Potential candidate residues engaging in stabilizing interactions are labelled and shown in stick representation. (B) Surface properties of the dimer interface and the periphery of RhsA. The surface of the interface is colored according to its Coulomb potential indicating positively (red, -20) and negatively (blue, +20) charged areas. The second representation shows the same interface but colored according

to hydrophobicity. Ochre indicates hydrophobic and white indicates hydrophilic regions. The area facing the outer peripheries of both barrels shows that these regions would electrostatically repel each other and explain why only two barrels can interact at the same time.
(TIF)

**S3 Fig. Characterization of the C-terminal autoproteolysis site of RhsA.** (A) Multiple sequence alignment of the C-terminal cleavage site. The cleavage site is indicated by the black arrow. Critical residues are highlighted by the residue numbering. Coloring is according to the ClustalW color code. (B) In *E. coli* toxicity assays show reduced toxicity of RhsA$_{\Delta TMD}$ harboring mutations of the catalytic aspartates D1324N and D1346N, respectively. Toxicity could by reversed by overexpression of the immune protein RhsI. The lower panel represents the same toxicity assay but with lower RhsA expression levels due to lower inducer concentration (L-rhamnose).
(TIF)

**S4 Fig. Local environment of the ordered toxin fragments inside the Rhs cocoon.** Interaction of three β-strands belonging to the toxin domain of RhsA with the Rhs barrel. The toxin fragments are stabilized by the interaction with hydrophobic and hydrophilic surfaces inside the Rhs barrel. The molecular surface is colored according to hydrophobicity with ochre and white indicating hydrophobic and hydrophilic regions, respectively. The electrostatic representation is colored according to Coulomb potential, which depicts positively (red, -20) and negatively (blue, +20) charged areas.
(TIF)

**S5 Fig. Characterization of the N-terminal autoproteolysis site of RhsA.** (A) The N-terminal cleavage site was identified by protein sequencing. The band containing the Rhs$_{cage}$ was excised and analyzed by *de novo* protein sequencing using LC-MS/MS. (B) Western blot analysis of wild-type or a N-terminal cleavage resistant mutant (H304A/P305A) confirming the location of the cleavage site at P305. (C) Multiple sequence alignment of the N-terminal cleavage site. The cleavage site is indicated by a black arrow. Coloring is according to the ClustalW color code. (D) Sequence alignment of the T6SS class I prePAAR effectors highlighting the N-terminal cleavage site. Residue conservation is depicted as Weblogo. The Weblogo represents residue conservation. (E) Cartoon representation of the N-terminal cleavage site in RhsA. Residues A312 and K313 as well as conserved and potential catalytically active residues are show in stick representation. Residues A312 and K313 correspond to the catalytically active glutamates in TseI [18]. (F) Hypothesized cysteine protease motif which is near the N-terminal cleavage site P305. (G) Multiple sequence alignment highlighting the conservation of the hypothesized cysteine protease motif among class I prePAAR effectors. The Weblogo represents residue conservation. (H) Western blot analysis of potential residues involved in N-terminal cleavage and generation of the cleavage product RhsA$_N$. Impaired cleavage was assessed by appearance of the full-length RhsA chain which is a mixture of both species, the C-terminally cleaved fragment (-CT) and the C-terminally uncleaved fragment (+CT). RhsA was coexpressed with its cognate chaperone EagR1. The blot was performed against N-terminal His$_6$-tagged proteins (α-His). (I) Western blot analysis to examine the potential involvement of RhsA's C-terminal toxin domain in the autoproteolysis of its N-terminal region. An RhsA truncation lacking the C-terminal toxin domain (RhsA$_{\Delta tox}$, residues 1–1323) was expressed and examined for N-terminal cleavage using antibodies that recognize its N-terminus (α-FLAG) or its Rhs cocoon (α-RhsA).
(TIF)

**S6 Fig. Hydrophilic interactions of the plug domain with the Rhs barrel.** (A) Only a few hydrophilic interactions stabilize the plug domain. Residues participating in hydrogen bonds are labeled and shown in stick representation. The cork and the seal are colored in orange whereas the Rhs repeats is colored in beige. (B) The barrel would not be closed without the observed density corresponding to the seal. The model for the seal was manually removed to visualize the opening though which toxin is threaded into the target cell after removal of the seal and the anchor helix.
(TIF)

**S7 Fig. Stabilization of the N-terminal seal and anchor helix of RhsA.** (A-B) A small density, corresponding to seal (residues K268-V275) of RhsA enters the barrel from the top (mesh). This results in a complete sealing of the cocoon. (C–E) The seal leads further down into an amphipathic helix which strongly interacts with the inner surface of the cocoon and thus serves as an anchor point for the N-terminal domain. The amphipathic helix is stabilized by hydrophobic interactions with Rhs repeats.
(TIF)

**S8 Fig. Comparison of Rhs repeat containing proteins of known structure.** (A) Comparison of the plug domains of RhsA (orange) with YenBC (PDB ID: 4IGL), TcdB2-TccC3 (PDB ID: 6H6G) and Ten2CT (PDB ID: 6FB3). (B) Sequence alignment of the plug domain of RhsA with the sequences of the homologous domains found in TcdB2-TccC3 (top) and YenBC (bottom) are shown. Residues are colored according to the ClustalW color code. (C) Structural overlay of RhsA with YenBC (left) and Ten2CT (right).
(TIF)

**S9 Fig. Cryo-EM of the PFC.** (A) Representative negative stain micrograph and 2D class averages of the intact PFC consisting of VgrG1, RhsA and EagR1. Scale bar micrograph, 100 nm. Scale bar 2D class averages, 10 nm. (B) Cryo-EM micrograph of the PFC and image processing workflow. Particles were picked with the general model of crYOLO and classified with ISAC. Refinement in MERIDIEN led to a reconstruction of 3.3 Å (FSC = 0.143 criterion). The final reconstruction was either colored individually for each protomer or according to local resolution. Fourier shell correlation was plotted according to two independent refined maps. Dashed line indicates the gold standard FSC criterion of 0.143. Scale bar micrograph, 100 nm. Scale bar 2D class averages, 10 nm. (C) Overlay of VgrG1 from *P. protegens* (green) with VgrG1 from *P. aeruginosa* (blue, PDB ID: 6H3N). (D) Map quality of selected parts of the structure (transparent surface) with built atomic models (cartoon representation). (E) Potential ion binding site in the middle part of the β-prism of *P. protegens* VgrG1. Sulfate ions were modelled into the spherical densities. Coordinating residues are displayed in stick representation.
(TIF)

**S1 Table. Bacterial strains used in this study.**
(DOCX)

**S2 Table. Plasmids used in this study.**
(DOCX)

**S1 Movie. Flexibility of the VgrG1-RhsA-EagR1 complex.** This video highlights the positional flexibility of RhsA relative to the VgrG1 spike protein.
(MP4)

**S1 Dataset. Dataset of *de novo* protein sequencing of RhsA.**
(ZIP)

## Acknowledgments

We thank D. Prumbaum and O. Hofnagel for assistance in electron microscopy data collection and maintaining the EM facility.

## Author Contributions

**Conceptualization:** John C. Whitney, Stefan Raunser.

**Data curation:** Dennis Quentin, Shehryar Ahmad, Kartik Sachar, John C. Whitney.

**Formal analysis:** Patrick Günther, Dennis Quentin, Shehryar Ahmad, Kartik Sachar, Christos Gatsogiannis.

**Funding acquisition:** John C. Whitney, Stefan Raunser.

**Investigation:** Patrick Günther, John C. Whitney.

**Project administration:** John C. Whitney, Stefan Raunser.

**Supervision:** John C. Whitney, Stefan Raunser.

**Validation:** Patrick Günther, Christos Gatsogiannis, Stefan Raunser.

**Visualization:** Patrick Günther, Shehryar Ahmad, Kartik Sachar.

**Writing – original draft:** Patrick Günther.

**Writing – review & editing:** Shehryar Ahmad, Kartik Sachar, John C. Whitney, Stefan Raunser.

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
