## [Decision Letter · Decision Letter 0]

24 Oct 2021

Dear Prof Raunser,

Thank you very much for submitting your manuscript "Structure of a bacterial Rhs effector exported by the type VI secretion system" for consideration at PLOS Pathogens. As with all papers reviewed by the journal, your manuscript was reviewed by members of the editorial board and by several independent reviewers. In light of the reviews (below this email), we would like to invite the resubmission of a significantly-revised version that takes into account the reviewers' comments.

All three reviewers reported the significance of the work and Reviewers #1 and #3 in addition commented on the well written manuscript and technical achievement. Reviewer #2 felt more functional work up was needed to support the model while Reviewer #1 specifically notes that additional studies could be part of a future studies.

The key data that seems to be missing relates to the N-terminal cleavage site and both Reviewer #2 and #3 offer different suggestions on how to resolve either using MS or by expressing a truncated constructs. It seems that these are reasonable experiments that could be done to clarify the exact location of the N-terminal processing.These concerns should be addressed with new experiments in a revised manuscript that includes  a text revision that addresses all other comments.

We cannot make any decision about publication until we have seen the revised manuscript and your response to the reviewers' comments. Your revised manuscript may be sent to reviewers for further evaluation.

Sincerely,

Karla J.F. Satchell, Ph.D.

Section Editor

PLOS Pathogens

Kasturi Haldar

Editor-in-Chief

PLOS Pathogens

orcid.org/0000-0001-5065-158X

Michael Malim

Editor-in-Chief

PLOS Pathogens

orcid.org/0000-0002-7699-2064

Reviewer's Responses to Questions

**Part I - Summary**

Reviewer #1: The manuscript by Günther et al. is a welcome addition to the field of type six secretion system (T6SS) effector biology. In this work, the authors report the high-resolution structure of the Pseudomonas protegens T6SS effector, RhsA, alone and in complex with its cognate spike protein, VgrG1. There is also an excellent comparison of all proteins containing Rhs elements of known structure (Tc toxins and teneurins). This additional comparison highlights how proteins with Rhs elements have been adapted to perform different biological functions and is an intriguing section of the manuscript (Line 239-265). The authors observe that the P. protegens RhsA forms a traditional Rhs cocoon-like shell that surrounds a C-terminal hypervariable region (HVR). As with other T6SS effectors, this HVR contains a toxin domain that mediates bacterial antagonism when delivered into a susceptible target bacterium. Autocatalytic cleavage of this toxin domain by a conserved aspartyl protease site is required for its release from the Rhs shell. The C-terminus of the Rhs shell is sealed shut in a similar manner to the Tc toxins, positioning the aspartyl protease into the interior of the shell. What is very interesting, and a novel finding, is how the RhsA shell is sealed shut at its N-terminal side. The authors observe that amino acids 268-386 form a “Plug” domain consisting of 3 parts: the Cork, Seal, and Anchor Helix. Through different hydrophobic and some hydrophilic interactions, the Cork fills in most of the space. This was nicely described as how a champagne cork would seal a bottle. The Cork is connected through a flexible region to an Anchor Helix that interacts with a specific section of the Rhs shell interior wall. The Anchor helix is then connected through another flexible region to the Seal, a peptide sequence that fills in a small gap in the Cork. They postulate that the Anchor Helix helps hold the Seal in place to close the shell. This would be an important adaptation, as the Anchor Helix and Seal are separated from the Cork between amino acids 304 and 305 by an unidentified N-terminal autocatalytic site. N-terminal processing of an Rhs protein has not been described in such detail and is an important finding for the field. Finally, the authors present the structure of RhsA, it’s cognate chaperone EagA, and its cognate spike protein VgrG1. Because of flexibility of the RhsA/EagA complex in orientation to VgrG1, the authors were unable to obtain a high-resolution structure of this entire complex but were able to present another structure of a VgrG spike. They were able to demonstrate that a VgrG trimer likely incorporates a phosphate or sulphate ion to stabilize the VgrG core. The authors put forth a model on how the RhsA toxin domain may be delivered into a susceptible targeted bacterium by the T6SS. This section relies on general knowledge of how T6SS firing events work, incorporation of studies on Tse6 (a different T6SS effector that contains a pre-PAAR motif and Trans-membrane helices), and findings from this work. While their model is sound and provides a nice conclusion to the manuscript, the exact process of toxin domain translocation across the inner membrane of a targeted bacterium has not been directly assessed for this protein; however, this aspect of the T6SS is outside the scope of this study. Overall, the manuscript is extremely well written, especially considering the technical and complex architecture of RhsA. All the figures are well designed and easy to interpret. The document was a joy to read, and I think is well suited for a broad audience. I did not observe any major concerns with this manuscript and feel that any additional experiments would be outside of the scope of this work. I did have a few minor points that I feel should be addressed prior to its final acceptance (discussed below). My congratulations to the authors for such excellent work.

Reviewer #2: The manuscript of Gunther and colleagues presents the structural analysis of type 6 secreted (T6S) effector RhsA, which shares sequence and structural features with Tc toxins and eukaryotic teneurin proteins. The large size of this protein allowed for the single particle cryo EM structure determination approach which resulted in 3.3 A resolution structure for the middle portion of the protein while the structures of N- and C-terminal portions of the protein remained mostly unresolved. The study also attempted the same structural approach for characterisation of “pre-firing” tri-protein complex of RhsA (T6S effector), EagR1 (T6S chaperon) and VgrG1 (T6S spike), which, unfortunately, did not provide on functionally important details on interactions between these proteins but provided a decent model for VgrG1 protein.

Based on this structural data the authors propose models for delivery of RhsA via interactions between N-terminal portion of the protein with T6SS machinery and for RhsA toxicity for the recipient cell via nuclease activity of C-terminal portion of the protein.

This study provides a partial molecular image of an important effector family, however it comes short of delivering significant mechanistical insight into the function of RhsA, which is expected from a molecular structure driven analysis.

This deficiency is primarily due to the lack of structural details in both N- and C-terminal portions in the presented RhsA structure, which does not allow for accurate prediction of specific elements involved in delivery via T6SS and toxic domain’s release into the cytoplasm of the recipient cell. While these shortcomings of structural data could have been compensated with an extensive functional analysis, presented functional data is very limited thus rendering proposed models of RhsA delivery and activity both over-speculative and imprecise.

To conclude, an extensive additional functional analysis addressing specific points discussed below will be needed to merit a positive review of this study.

Reviewer #3: The paper by Günther et al. describes the first structure of an RhsA-type effector exported by the Type VI secretion system (T6SS). RHS (rearrangement hotspot repeat) containing proteins were first identified in E. coli in the 1980s and for many years remained enigmatic. The function of the RHS repeat sequence in producing a cocoon-shaped protein chaperone was identified in 2013 with the structural characterisation of the BC subcomplex of the tripartite ABC toxin complexes (Tc), and around the same time, homologues of the prototypic E. coli RhsA protein were identified as being mediators of contact-dependent inhibition via the T6SS. However, the mechanistic details of how RhsA proteins function as T6SS effectors has not be clear. Hence the structure of RhsA from Pseudomonas protegens described here constitutes a significant advance in our understanding of these systems. The paper is also a technical achievement, as RhsA is at the lower end of molecular size readily visualisable by single-particle analysis (although this is mitigated by its dimerisation) and is composed entirely of β-sheet structure, which provides challenges to particle alignment.

The architecture of the RHS-repeat region, protease and C-terminal toxin regions was predictable by comparison with the BC proteins, but a major finding in this paper is the arrangement of the N-terminal region of the protein. The N-terminal domain acts as a plug capping the RHS cocoon, and unexpectedly shows a weak structural similarity (and previously undetected sequence similarity) to the equivalent domain in TcB.

The observed dimerisation of RhsA is striking on two counts. The head-to-tail dimers are initially striking for their similarity to dimers of the eukaryotic RHS-repeat containing protein Teneurin. It is also striking that this arrangement seems non-productive in the context of the T6SS. This is an issue that the authors should address. Presumably the dimer is an artefact of the isolated VgrG1-RhsA complex? Is it possible to rationalise the structure of the VgrG1-RhsA complex seen here with the T6SS spike structure reported by Nazarov et al. (doi.org/10.15252/embj.201797103) ?

Although it builds on previously published work, the delivery model suggested in Figure 6 is the most speculative part of the paper, and does not seem fully supported by the available data. It is unclear how two trans-membrane helices will be sufficient to form a pore able to translocate a polypeptide through the inner membrane. Is it possible that inner membrane proteins in the target cell are also required? It is additionally unclear how the T6SS contractile event does not simply push the sharp PAAR tip through the inner membrane. Not all PAAR-containing effectors contain a TM-helix-containing pre-PAAR domain, which implies at least the existence of an alternate mechanism.

**Part II – Major Issues: Key Experiments Required for Acceptance**

Reviewer #1: None

Reviewer #2: Both N- and C-terminal portions of RhsA effector undergo autocleavage as have been shown for other members of this family (Pei et al., 2020). Current study attempts to use obtained RhsA structure to gain insight into both these processes but fails to deliver. This shortcoming should be addressed via appropriate functional assays.

The mechanism of C-terminal portion cleavage appears to be similar to the process described for the Tc toxins and previously characterised representatives of this family. The authors suggest that the C-terminal portion cleavage can be abrogated via mutations of conserved catalytic residues, that are validated by the competition assay (Fig 2C). However, the overexpression of the same mutants in the assay presented at Supp Fig 3E suggests that these mutants retain some (less than the wild type) toxicity. Why isn’t this toxicity manifesting in the competition assay? The biochemical activity of C-terminal toxin is only implied via sequence similarity but never tested. Is the activity of C-terminal domain inhibited by the appropriate immunity protein? The presence of immunity protein often helps stabilizing and obtaining the structure of toxic domain. Was this attempted?

Presented effort for gaining insight into RhsA’s N-terminal portion’s cleavage is even less convincing. The specific position of the cleavage is determined based on the absence of interpretable density after residue 305, which is not particularly accurate and can also be due to lability of this portion of the structure. There are specific mass spectrometry based assays that should be used to identify/confirm the specific position of N-terminal cleavage. Wrongly estimated cleavage site could be one of possible reasons for failing to identify the catalytic residues responsible for auto-proteolysis of N-terminal portion. Obtaining non cleavable version of RhsA could have provided insights into this mechanism. Since the mutation of “usual suspects” did not reveal the culprits more unbiased approach (alanine scanning mutagenesis?) must be applied.

The model proposed for conformational changes of RhsA upon delivery by T6SS into recipient cell (Fig. 6) is by far the weakest and most over-speculative part of this study. Primarily based on experimentally unconfirmed position of the cleavage site it liberally uses analogy with Tc toxin while continuously highlighting the significant differences between these proteins and RhsA. Since no structural or functional data is provided in support of this model, I see no way of assessing its merit. Just to provide a specific example, provided model suggests that the 2 predicted transmembrane helices will facilitate the threading and/or translocation of either toxin or the entire cocoon/toxin portion through the inner membrane into recipient cell’s cytoplasm. Would authors provide an example of a translocation channel composed with only 2TMs that their model is based on? Any experimental assays tried to show the formation of such translocation channel by these 2 TMs?

Reviewer #3: The autoproteolysis of the N-terminal domain is a significant new finding. Frustratingly, mutation of the identified possible active site residues for the protease activity (e.g. C538) does not alter protease activity. It is therefore suggested that the encapsulated toxin contributes active site residues for the protease activity. Given that the C-terminal toxin domain is unfolded within the cocoon, it seems unlikely that it can contribute active site residues to the proteolytic cleavage. However, this idea is testable: Is the N-terminal domain cleaved if a truncated protein lacking the C-terminal toxin is expressed? This is a relatively straightforward experiment that should be performed.

**Part III – Minor Issues: Editorial and Data Presentation Modifications**

Reviewer #1: 1. Figure 2C, y-axis: I just want to clarify that this is the Log10 of the Competitive Index as calculated in the methods, i.e. Log10((Donor CFU/Recipient CFU)endpoint / (Donor CFU/Recipient CFU)beginning). Perhaps just clarify in the methods.

2. Figure 2G, RhsA∆N: I could not find a exact definition of this construct. Based on the main text line 169, I’m assuming it is an N-terminal deletion of RhsA at AA305, but I wound define this either in the legend or in the main text.

3. Supplementary Figure 3C: Should residue 313 be a K not an A?

4. Supplementary Figure 3H: There is also impaired cleavage of the H530A construct, but this was not mentioned in the text. This residue is highlighted in Supplementary Figure 3G, and thus seems like an important residue to comment on. This construct also appears to have a majority of -CT bands, not +CT bands, where the D318A/D319A/D320A construct has a mixture of both. Is there a possible relation of this N-terminal region to C-terminal cleavage? This may be an instance where further experiments are outside the scope of this manuscript, but I think it may be worth a comment.

5. Sentences from line 333 to 338. I think this appears to be the most plausible scenario. What is not hypothesized here is how the N-terminal TM helices, which have been cleaved from the Rhs shell, may subsequently interact with the C-terminal toxin as it leaves the Rhs shell through the opened seal. Would you propose that the Shell and TM helices come together to funnel the toxin across the inner membrane, or do you see release of the toxin from the Rhs shell as a separate event from membrane translocation (i.e. the toxin has to “find” the TM helices to translocate across the inner membrane). I understand this section is a proposed model, but it may be strengthened by a bit more explanation.

Reviewer #2: Given the important similarity between RhsA and Tc toxins/ teneurin discussed throughout the manuscript more information about these proteins must be provided in the introduction.

L-62-63: “latter likely plays a role in target in cell penetration”. In the context of this sentence “latter” refers to transmembrane helices while the cell penetration is rather function of VgrG spike, no? Consider rewording.

L-74: “Tc toxin”. – As per my general comment above the Tc toxin needs proper introduction/explanation prior to this statement.

L-134: Supp Fig 3C-E. Should be corrected to Supp Fig 3B.

L-151: Is the functional prediction based on structural similarity or primary sequence features?

L-246-249: “hyperconserved region”. I am not sure there is such a term. Earlier in introduction, C-terminal part of Rhs is mentioned as “hypervariable region”, but within this domain there is hyperconserved region?

Fig 3C: Is it teneurin structure presented here? If so, please add the proper statement to the legend as the title for Fig 3 refers to RhsA structure. From main text, it appears to be the teneurin structure

L-309: “After cleavage…” Is it after cleavage that EagR1 binds to N-terminal TM or is it before? From Fig6, it looks like EagR1 binds before the cleavage.

L-327: “As proposed previously…” Appropriate reference is needed.

Reviewer #3: Line 79: It’s not clear what is meant by “host cell targeting”.

Line 93 and elsewhere: RHS repeats do not form a β-barrel, which is a specific defined protein topology where β-sheets form a closed toroid distinguished by continuous hydrogen bonds between strands. In contrast RHS repeats form an extended β-sheet which is folded around onto itself. This is best described as a “cocoon “or a “shell” instead.

Line 168-169 -ΔN and ΔNT is somewhat confusing terminology

Line 226: The phrase “Rhs core” has been used previously in the published literature to refer to hyperconserved autoprotease region. Here the cocoon is referred to as the “core”, which is potentially confusing.

Lines 752-756 Should the PDB and EMDB IDs be listed?

PLOS authors have the option to publish the peer review history of their article (what does this mean?). If published, this will include your full peer review and any attached files.

Reviewer #1: No

Reviewer #2: No

Reviewer #3: **Yes: **J. Shaun Lott
---

## [Editor Report · Decision Letter 1]

8 Dec 2021

Dear Prof Raunser,

We are pleased to inform you that your manuscript 'Structure of a bacterial Rhs effector exported by the type VI secretion system' has been provisionally accepted for publication in PLOS Pathogens.

Best regards,

Karla J.F. Satchell, Ph.D.

Section Editor

PLOS Pathogens

Karla Satchell

Section Editor

PLOS Pathogens

Kasturi Haldar

Editor-in-Chief

PLOS Pathogens

orcid.org/0000-0001-5065-158X

Michael Malim

Editor-in-Chief

PLOS Pathogens

orcid.org/0000-0002-7699-2064
---

## [Editor Report · Acceptance letter]

31 Dec 2021

Dear Prof Raunser,

We are delighted to inform you that your manuscript, "Structure of a bacterial Rhs effector exported by the type VI secretion system," has been formally accepted for publication in PLOS Pathogens.

Best regards,

Kasturi Haldar

Editor-in-Chief

PLOS Pathogens

orcid.org/0000-0001-5065-158X

Michael Malim

Editor-in-Chief

PLOS Pathogens

orcid.org/0000-0002-7699-2064